# Constitutive hydrogen inhalation prevents vascular remodeling via reduction of oxidative stress

**Takeshi Kiyoi**[1,2], **Shuang Liu**[1], **Erika Takemasa**[1], **Hirotomo Nakaoka**[3], **Naohito Hato**[4], **Masaki Mogi**[1] *

**1** Department of Pharmacology, Ehime University Graduate School of Medicine, Tohon, Ehime, Japan,
**2** Division of Analytical Bio-medicine, Advanced Research Support Center, Ehime University, Toon, Ehime, Japan, **3** Division of Laboratory Animal Research, Advanced Research Support Center, Ehime University, Toon, Ehime, Japan, **4** Department of Otorhinolaryngology, Head and Neck Surgery, Ehime University Graduate School of Medicine, Tohon, Ehime, Japan

* mmogi@m.ehime-u.ac.jp

**Data Availability Statement:** All relevant data are within the manuscript and its Supporting Information files.

## Abstract

Molecular hydrogen is thought to have an inhibitory effect on oxidative stress, thereby attenuating the onset and progression of various diseases including cardiovascular disease; however, few reports have assessed the preventive effect of constitutive inhalation of hydrogen gas on of vascular remodeling. Here, we investigated the effect of constitutive inhalation of hydrogen gas on vascular neointima formation using a cuff-induced vascular injury mouse model. After constitutive inhalation of compressed hydrogen gas (O2 21%, N2 77.7%, hydrogen 1.3%) or compressed air only ($O_2$ 21%, $N_2$ 79%) by C57BL/6 mice for 2 weeks from 8 weeks of age in a closed chamber, inflammatory cuff injury was induced by polyethylene cuff placement around the femoral artery under anesthesia, and hydrogen gas administration was continued until sampling of the femoral artery. Neointima formation, accompanied by an increase in cell proliferation, was significantly attenuated in the hydrogen group compared with the control group. NADPH oxidase NOX1 downregulation in response to cuff injury was shown in the hydrogen group, but the expression levels of NADPH oxidase subunits, p40phox and p47phox, did not differ significantly between the hydrogen and control groups. Although the increase in superoxide anion production did not significantly differ between the hydrogen and control groups, DNA damage was decreased as a result of reduction of reactive oxygen species such as hydroxyl radical ($\cdot$OH) and peroxynitrite ($ONOO^-$) in the hydrogen group. These results demonstrate that constitutive inhalation of hydrogen gas attenuates vascular remodeling partly via reduction of oxidative stress, suggesting that constitutive inhalation of hydrogen gas at a safe concentration in the living environment could be an effective strategy for prevention of vascular diseases such as atherosclerosis.

**Funding:** This study was supported by JSPS KAKENHI Grant Numbers 18K08389 (S.L.) and 25462220 (M.M.), and research grants from Panasonic Corporation (https://www.panasonic.com/jp/home.html). The funders had no role in the study design, data collection and analysis, decision to publish, or preparation of the manuscript.

**Competing interests:** The authors have declared that no competing interests exist.

**Abbreviations:** CVD, cardiovascular disease; DHE, dihydroethidium; ERK, extracellular signal-regulated kinase; HPF, hydroxyphenyl fluorescein; JNK, c-Jun NH2-terminal kinase; MAPK, mitogen-activated protein kinase; NADPH, nicotinamide adenine dinucleotide phosphate; NOX1, NADPH oxidase 1; PCNA, proliferating cell nuclear antigen; ROS, reactive oxygen species; VSMC, vascular smooth muscle cell.

## Introduction

Cardiovascular disease (CVD), including ischemic heart disease, remains the leading cause of health loss and death worldwide [1]. It has been suggested that lifestyle-related diseases such as hypertension, diabetes and obesity are involved in the onset of CVD, and disease progression is accompanied by vascular injury due to reactive oxygen species (ROS)-dependent chronic/persistent oxidative stress [2, 3]. Oxidative stress refers to elevated levels of intracellular ROS, which cause damage to lipids, proteins and DNA [4]. Therefore, reduction of oxidative stress by down-regulating ROS could be an approach for prevention of the onset of CVD. Indeed, it was reported that angiotensin II receptor blockers attenuate atherosclerosis as a result of down-regulating ROS by inhibition of nicotinamide adenine dinucleotide phosphate (NADPH) oxidase activity [5].

It is reported that molecular hydrogen attenuates oxidative stress by acting as a radical scavenger for hydroxyl radical ($\cdot$OH) and peroxynitrite (ONOO-) in vitro [6]. Since then, molecular hydrogen has been proved to bring about beneficial effects on the pathophysiology of various diseases through reduction of oxidative stress [7–10]. There are several convenient and effective delivery systems such as inhalation, oral intake of hydrogen-rich water, injection of hydrogen-rich saline and direct incorporation (bath, eye drops, etc.) for molecular hydrogen administration in vivo [11]. It was suggested that molecular hydrogen prevents vascular remodeling in animal models such as ischemia and reperfusion (I/R) injury, vein grafting, carotid balloon injury and cerebral vasospasm of subarachnoid hemorrhage via reduction of oxidative stress [12–15].

In previous reports, 2% hydrogen gas inhalation during a 2-hour ischemic condition before reperfusion was found to be effective for mitigation of mortality and functional outcome in a rat I/R injury model [12]. However, little is known about the beneficial effects of constitutive inhalation of hydrogen gas on the prevention of CVD in daily living. Recently, hydrogen gas, which can be easily produced from water by electrolysis, has not only received attention as an energy source, but is also expected to contribute to a healthy lifestyle. Therefore, we investigated the effect of constitutive administration of hydrogen gas at a low concentration on vascular remodeling using a cuff-induced vascular injury model. In this study, we focused on the effects of hydrogen gas inhalation on CVD as a lifestyle intervention. CVD is induced by lifestyle-related disease with chronic/persistent oxidative stress; that is, the constitutive inhalation of molecular hydrogen in real life contributes to reducing chronic/persistent oxidative stress and has the potential to prevent CVD.

## Materials and methods

### Animals and treatment

C57BL/6 mice were purchased from CLEA Japan, Inc. (Tokyo, Japan). Male mice aged 8 weeks were used for all experiments (median weight 23 g). Eight animals were housed in 350 mm x150 mm x 150 mm closed chambers as shown in **Fig 1**. The mice were randomly assigned to the hydrogen or control group. Compressed hydrogen gas ($O_2$ 21%, $N_2$ 77.7%, hydrogen 1.3%) or compressed air ($O_2$ 21%, $N_2$ 79%) flowed continuously at 0.4 L/min. Humidity (up to 70%) and temperature (around 25˚C) in the closed chamber were monitored and maintained using dehumidifiers and deodorants. Animal bedding was changed every two days. Rooms were kept at a constant temperature of 25˚C, with an automatically controlled 12:12 h light-dark cycle with lights on at 7:00 a.m. Food and water were provided ad libitum.

After inhalation of 1.3% hydrogen gas for 2 weeks from 8 weeks of age, inflammatory cuff injury was induced by polyethylene cuff placement around the femoral artery under anesthesia

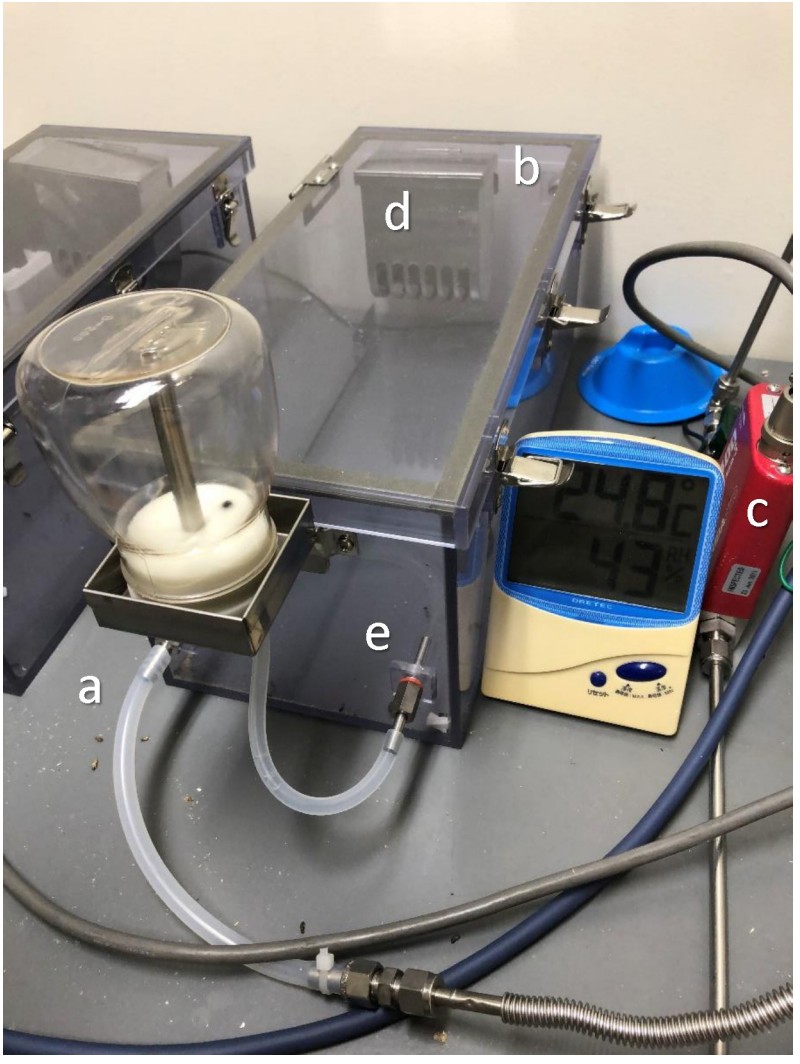

**Fig 1. A Closed chamber used in our experiments.** (a) Gas inlet, (b) gas outlet, (c) gas flow controller, (d) bait box, and (e) feed-water inlet.

as described previously [16–18]. All mice underwent cuff placement surgery within an hour after administration of 0.1 ml/10 g of combination anesthetic (0.3 mg/kg medetomidine, 4.0 mg/kg midazolam, 5.0 mg/kg butorphanol) in saline by intraperitoneal injection. To sample the femoral artery in each experiment, blood-letting was performed by transcardial perfusion with PBS for euthanasia under anesthesia as described above. All animal experiments were conducted between 9:00 a.m. and 18:00 p.m. in our laboratory.

Experimental protocols were in accordance with the guidelines of the Animal Care Committee of Ehime University and approved by the University Committee for Animal Research.

## Morphometric analysis

In morphometric analysis, the femoral arteries, which had undergone cuff placement for 14 days under inhalation of 1.3% hydrogen gas, were taken after perfusion with 4% paraformaldehyde (PFA), and rinsed for 24 hours with 4% PFA at 4˚C. Serial 4-μm paraffin-embedded sections were stained with Elastica van Gieson (EVG) for the observation of neointima

formation. The sections were observed under a light microscope BZ9000 (Keyence, Osaka, Japan), and neointimal area was measured using ImageJ software.

### Immunohistochemical staining

Paraffin-embedded samples for morphometric analysis were also used for immunohistochemical study. Sections were treated with 3% $H_2O_2$ for 10 min to block endogenous peroxidase, and antigen retrieval was performed by heat treatment with citrate buffer solution (pH 6.0). Sections were incubated overnight at 4˚C with the primary antibody, proliferating cell nuclear antigen (PCNA) antibody (Abcam, Ltd., Newcastle upon Tyne, UK). Antibody binding was visualized with 3, 3′-diaminobenzidine (DAB) using a mouse staining kit, Histofine (Nichirei Bioscience, Tokyo, Japan). The staining results were evaluated by counting the number of PCNA-positive cells in the neointima.

### Laser microdissection

Femoral arteries that had undergone cuff placement for 7 days under inhalation of 1.3% hydrogen gas were taken, and non-fixed frozen sections on foil-covered slides (Leica Microsystems, Wetzlar, Germany) were prepared. Sections were fixed with ethanol containing 5% acetic acid and stained with toluidine blue. Neointima and arterial media tissues were collected from femoral artery tissues by laser microdissection method using a LMD7000 (Leica Microsystems, Wetzlar, Germany). Each specimen was gathered in a 0.2 ml tube.

### RT-PCR

Specimens obtained using laser microdissection or pooled samples of 8–10 arteries for the group without cuff placement and 4–6 arteries for the group at 7 days after cuff placement were used. Total RNA was extracted from the femoral arteries using Sepasol RNA I Super G (Nacalai Tesque, Kyoto, Japan). Expression of mRNA was quantified by SYBR Premix Ex Taq using a Thermal Cycler Dice Realtime System (Takara Bio, Shiga, Japan). The sequences of PCR primers are given in **S1 Table**.

### Measurement of Reactive Oxygen Species (ROS)

Femoral arteries that had undergone cuff placement for 7 days under inhalation of 1.3% hydrogen gas were taken, and serial 6-μm non-fixed frozen sections were prepared. For detection of superoxide anions ($O_2^-\cdot$), 10 μmol/L fluorogenic dihydroethidium (DHE) (Abcam, Ltd., Newcastle upon Tyne, UK) was added, and sections were incubated for 30 min at 37˚C. The method was described previously [16, 17]. For detection of hydroxyl radicals (·OH) and peroxynitrite ($ONOO^-$), 10 μmol/L hydroxyphenyl fluorescein (HPF) (GORYO Chemical, Inc., Sapporo, Japan) was added and sections were incubated for 30 min at 37˚C. The HPF staining method was described previously [19]. The results were obtained using a fluorescence microscope BZ9000 (Keyence, Osaka, Japan). Intensity of fluorescence in the neointima and arterial media was analyzed and quantified using ImageJ software.

### Analysis of DNA damage

Frozen samples for measurement of ROS were used for analysis of DNA damage by detection of 8-nitroguanine and 8-hydroxy-2'-deoxyguanosine (8-OHdG). Serial 6-μm non-fixed frozen sections were prepared. Some sections were fixed for 5 min with ethanol. The sections were incubated overnight at 4˚C with anti-8-nitroguanosine rabbit polyclonal primary antibody (Dojindo Molecular Technologies, Inc., Kumamoto, Japan) or anti-8-OHdG rabbit polyclonal

primary antibody (Bioss Antibodies Inc., Woburn, MA), followed by an anti-rabbit secondary antibody (Thermo Fisher Scientific K.K., Waltham, MA). Antibody binding was visualized by Alexa 594 using a fluorescence microscope BZ9000. Intensity of fluorescence in the neointima and arterial media was analyzed and quantified using ImageJ software.

### Statistical analysis

All values are expressed as mean ± S.D. in the figures. Data were evaluated by ANOVA. If a statistically significant effect was found, post hoc analysis was performed to detect the difference between the groups. Values of P < 0.05 were considered statistically significant.

## Results

### Inhibitory effect of hydrogen inhalation on neointima formation

We examined the effect of hydrogen gas on neointima formation, 14 days after polyethylene cuff placement around the femoral artery. Neointima formation was observed in the hydrogen and air groups. The ratio of neointimal area to vascular media area in the hydrogen group (Hyd) was significantly attenuated 0.55-fold compared with that in the control air group (Con) (**Fig 2**).

### Inhibitory effect of hydrogen inhalation on cell proliferation

We observed that hydrogen gas decreased the neointimal area, with a decrease in PCNA labeling index in the intima. PCNA-positive cells in the neointima were observed in both the hydrogen and air groups. PCNA labeling index in the neointima in Hyd was decreased 0.69-fold compared with that in Con (**Fig 3**). In addition, most of the cells that proliferated in the intima after cuff placement were α-SMA positive (**S1 Fig**).

### Inhibitory effect of hydrogen inhalation on expression of NADPH oxidase subunits

We assessed the effect of hydrogen gas on mRNA levels of NOX1 (a type of NADPH oxidase) and NOX1 subunits (p40phox, p47phox) in the neointima and arterial media of the femoral artery 7 days after cuff placement. NOX1 expression level was significantly reduced in Hyd, but p40phox and p47phox expression levels did not differ significantly between Con and Hyd. NOX1 expression level in Hyd was 0.23-fold compared with that in Con (**Fig 4**).

### Inhibitory effect of hydrogen inhalation on ROS production

We examined the effect of hydrogen gas on production of ROS such as superoxide anion, hydroxyl radicals and peroxynitrite in the neointima and arterial media of the femoral artery 7 days after cuff placement. Production of superoxide anion was evaluated by DHE staining, and production of hydroxyl radicals and peroxynitrite was evaluated by HPF staining. There was no significant difference in production of superoxide anion between Con and Hyd (**Fig 5A**). On the other hand, hydroxyl radicals and peroxynitrite were markedly attenuated 0.84-fold in Hyd compared with those in Con (**Fig 5B**).

### Inhibitory effect of hydrogen inhalation on DNA damage by ROS

We investigated the effect of hydrogen gas on ROS-induced DNA damage in the neointima and arterial media of the injured artery 7 days after cuff placement. Nitroguanosine is a marker of DNA damage by peroxynitrite (ONOO⁻) and hydrogen peroxide, and 8-OHdG is also a

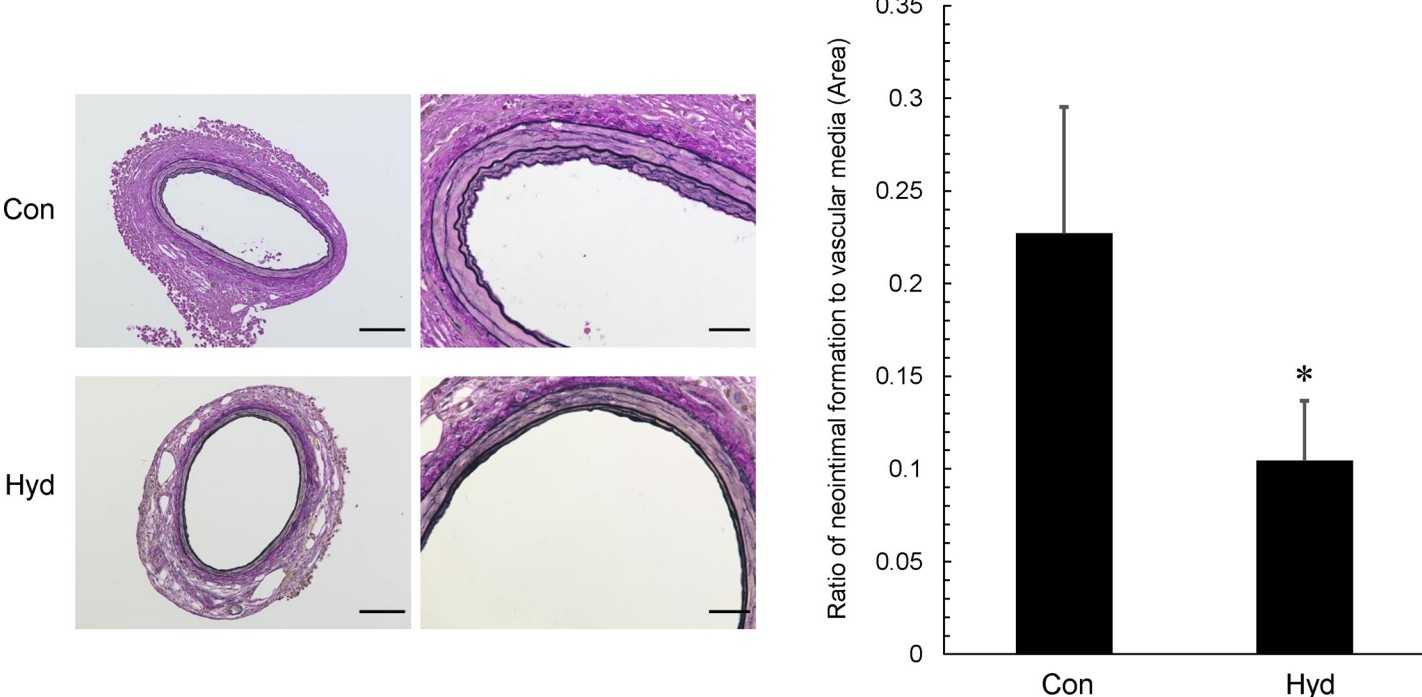

**Fig 2. Effect of hydrogen gas inhalation on neointima formation in injured femoral artery after cuff placement.** After inhalation of hydrogen gas for 2 weeks from 8 weeks of age in C57BL/6 mice, cuff injury was induced by polyethylene cuff placement around the femoral artery. Representative photos and quantitative analysis of neointimal area in cross sections of femoral artery with elastic van Gieson (EVG) staining are shown. Original magnification ×200 (scale bar: 100 μm) and ×600 (scale bar: 30 μm). In quantitative analysis, data represent the ratio of neointima formation area to vascular media area, and values are mean ± SEM (n = 16 for control group (Con), n = 24 for hydrogen group (Hyd)). *p<0.05 vs. Con.

representative marker of DNA damage by hydroxyl radicals (·OH). DNA damage by ROS was determined by immunocytochemical staining. Fluorescence labeling index of 8-nitroguanosine was significantly reduced 0.93-fold in Hyd compared with that in Con (**Fig 6A**). Fluorescence labeling index of 8-OHdG was also significantly reduced 0.84-fold in Hyd compared with that in Con (**Fig 6B**).

## Discussion

These results demonstrated that constitutive inhalation of hydrogen gas at a low concentration attenuated vascular remodeling via reduction of oxidative stress and proliferative signaling. Previous reports on the effects of hydrogen gas inhalation on CVD have focused on clinical application using a rat I/R injury model [12, 20]. On the other hand, this study on the effects of hydrogen gas inhalation on CVD focused on lifestyle intervention. In the present study, considering the daily life of humans, the preventive effect of constitutive administration of hydrogen gas on vascular remodeling was examined in a mouse cuff injury model.

In this vascular injury model, it is known that superoxide resulting from increased NADPH oxidase activity promotes VSMC proliferation and neointima formation [21]. NADPH oxidase subunits (p40phox and p47phox) are also known to promote vascular remodeling including atherosclerosis as potent positive regulators [22, 23]. Zhang et al. indicated that intraperitoneal injection of hydrogen-rich medium produced a decrease in expression of NADPH oxidase in an isoproterenol (ISO)-induced cardiac hypertrophy model rat [24]. Qin et al. reported that injection of hydrogen-rich saline reduced superoxide and prevented VSMC proliferation and migration in a rat carotid balloon injury model [13]. These reports suggest that molecular

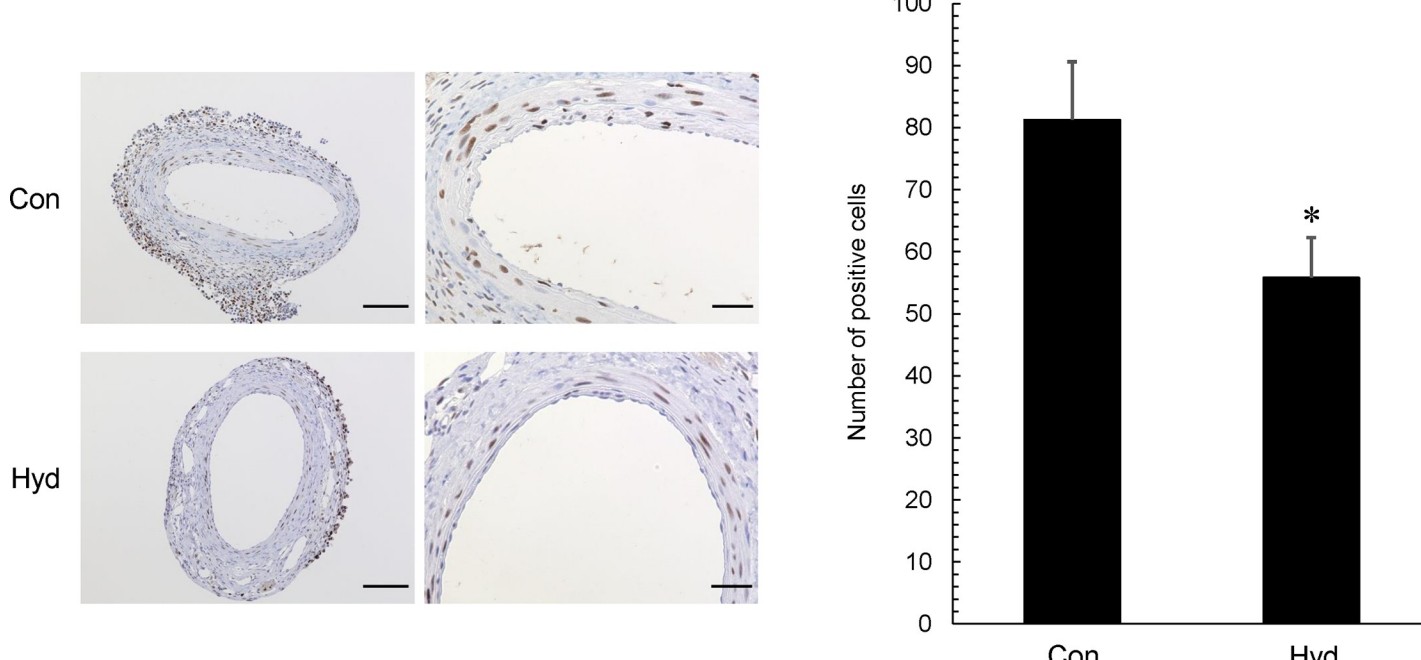

**Fig 3. Effect of hydrogen gas inhalation on cell proliferation in injured femoral artery after cuff placement.** Representative photos and quantitative analysis of injured femoral artery in cross-sections after immunohistochemical staining (for PCNA) are shown. Original magnification ×200 (scale bar: 100 μm) and ×600 (scale bar: 30 μm). In quantitative analysis, data represent the number of PCNA-positive cells in neointima and vascular media, and values are mean ± SEM (n = 16 for each group). *p<0.05 vs. Con.

hydrogen can prevent CVD by downregulating NADPH activity and superoxide production. However, molecular hydrogen does not act as a radical scavenger for other ROS such as superoxide, hydrogen peroxide etc. in vitro [6]. This is an unclear point regarding the mechanism of action of molecular hydrogen on biological activity. In the present results, inhalation of hydrogen gas downregulated the expression of NADPH oxidase, NOX1, but did not affect the expression levels of NADPH oxidase subunits such as p40phox and p47phox in the femoral artery 7 days after cuff placement. In addition, there was no significant difference in superoxide production between the hydrogen and control groups. These results suggest that our hydrogen administration system may have little effect on superoxide production via NADPH oxidase activity because there was no significant change in expression levels of NADPH oxidase subunits. On the other hand, our results were in agreement with the fact that molecular hydrogen does not directly act as a radical scavenger against superoxide [6].

Superoxide generation is the first step in the pathway of generating various ROS such as hydrogen peroxide, hydroxyl radicals (·OH) and peroxynitrite (ONOO⁻) [21,22]. Oxidative stress in CVD is not only affected by superoxide, but also by ·OH and ONOO⁻, which have high oxidant reactivity [3, 25]. Molecular hydrogen alleviates oxidative stress by acting as a radical scavenger for ·OH and ONOO⁻ in vitro [6]. In addition, some reports suggest that molecular hydrogen has an inhibitory effect on oxidative stress-mediated disease through a decrease in ·OH and ONOO⁻ levels in vivo [26, 27]. Igarashi et al. reported that hydrogen prevents corneal endothelial damage in phacoemulsification cataract surgery through reduction of ·OH [27]. Zhang et al. reported that drinking hydrogen-rich water markedly inhibited the formation of ONOO⁻ based on detection of 3-nitrotyrosine in abdominal arteries above and close to the coarctation site in a rat abdominal aortic coarctation model [26]. On the other

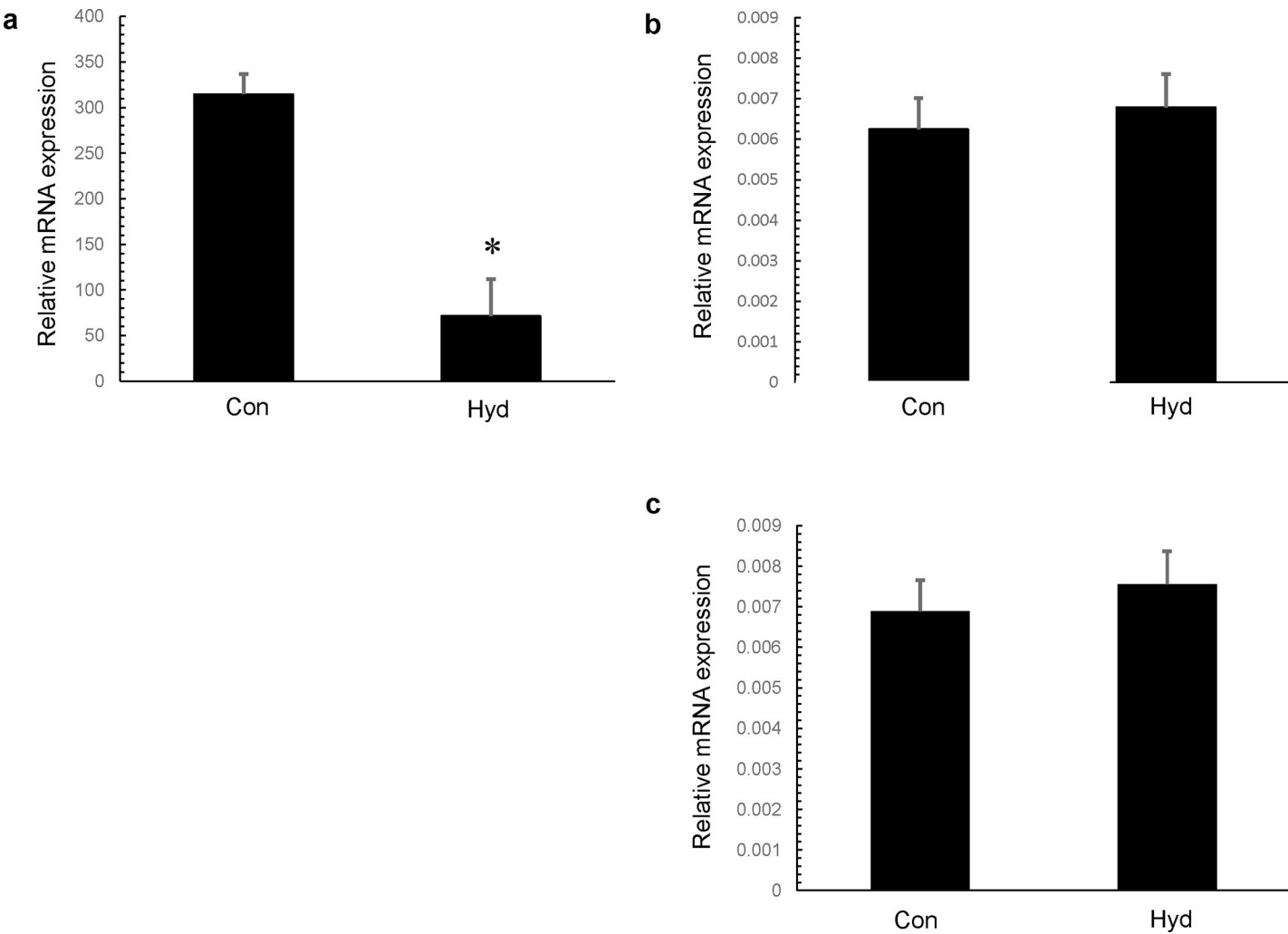

**Fig 4. Effect of hydrogen gas inhalation on NADPH oxidase and NADPH oxidase subunits.** Expression of NOX1 (a), p40phox (b) and p47phox (c) determined by real-time quantitative RT-PCR in femoral artery 7 days after cuff placement. Tissue samples were prepared from cuffed arteries 7 days after operation. Values are mean ± SEM (n = 5 for each group). Con; control group, Hyd; hydrogen group.

hand, it has also been pointed out that sufficient hydrogen is needed for a decrease in ·OH level [8]. In our experiment, there was a marked difference in attenuation of ·OH or, alternatively, ONOO⁻ level in the femoral artery 7 days after cuff placement between the hydrogen and control groups. Therefore, the results suggested that our hydrogen administration system might contribute to partial alleviation of ROS-dependent oxidative stress.

ROS-induced DNA damage and subsequent repair pathways are now increasingly appreciated as a risk factor for disease progression in CVD, including atherosclerosis [28]. Especially, ROS-induced DNA damage is strongly influenced by ·OH and ONOO⁻ because of strong oxidation. It is well known that 8-hydroxydeoxyguanosine (8-OHdG) is a representative marker of ROS-induced DNA damage [29, 30], and 8-nitroguanosine is also appreciated as a marker of DNA damage by ROS such as ONOO⁻ [31, 32]. In our experiment, both 8-OHdG and 8-nitroguanosine detection levels were decreased in the femoral artery 7 days after cuff placement in the hydrogen group. These results suggest that our hydrogen administration system

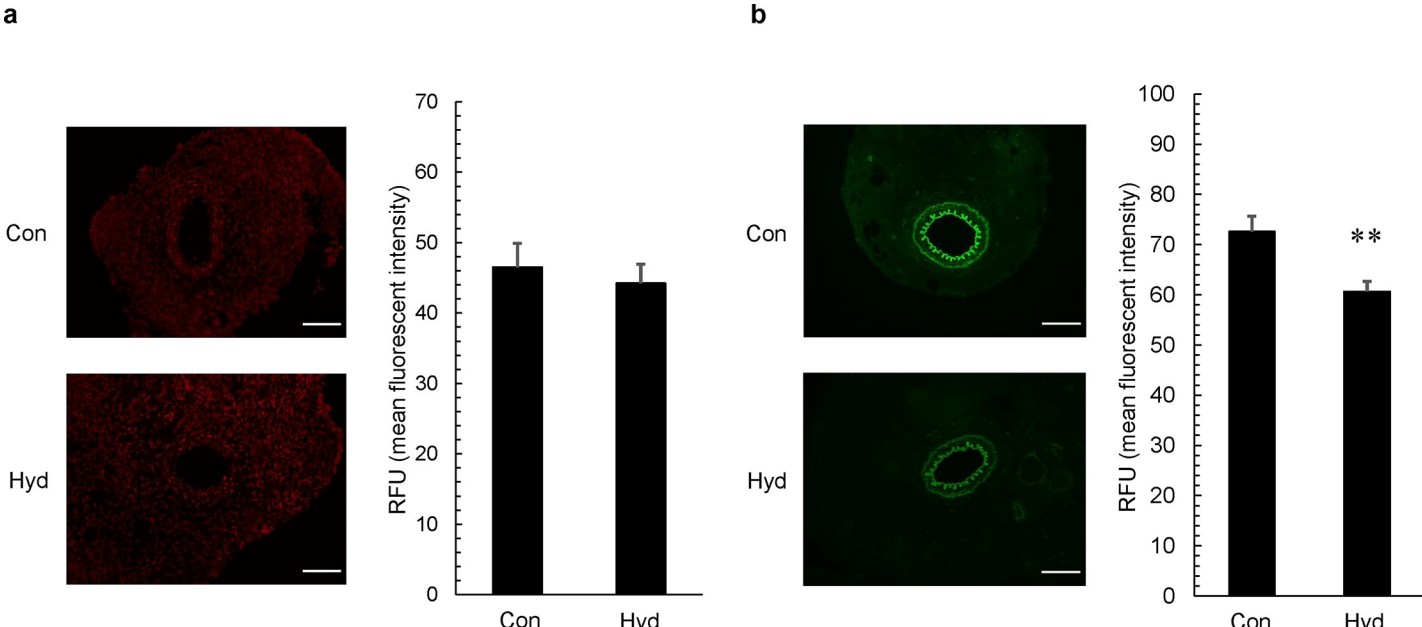

**Fig 5. Effect of hydrogen gas inhalation on ROS production.** Tissue samples were prepared from cuffed arteries 7 days after operation. (A) Representative photos of cross-sections of injured femoral artery after DHE staining and fluorescence intensity in intima and media. (B) Representative photos of cross-sections of injured femoral artery after HPF staining and fluorescence intensity in intima and media. The original photos were obtained as 8-bit images (original magnification ×200; scale bar: 100 μm), and data represent relative fluorescence units (RFU). Values are mean ±SEM (n = 18 to 21 for each group). $**p<0.01$ vs. air group (Con). Hyd; hydrogen group.

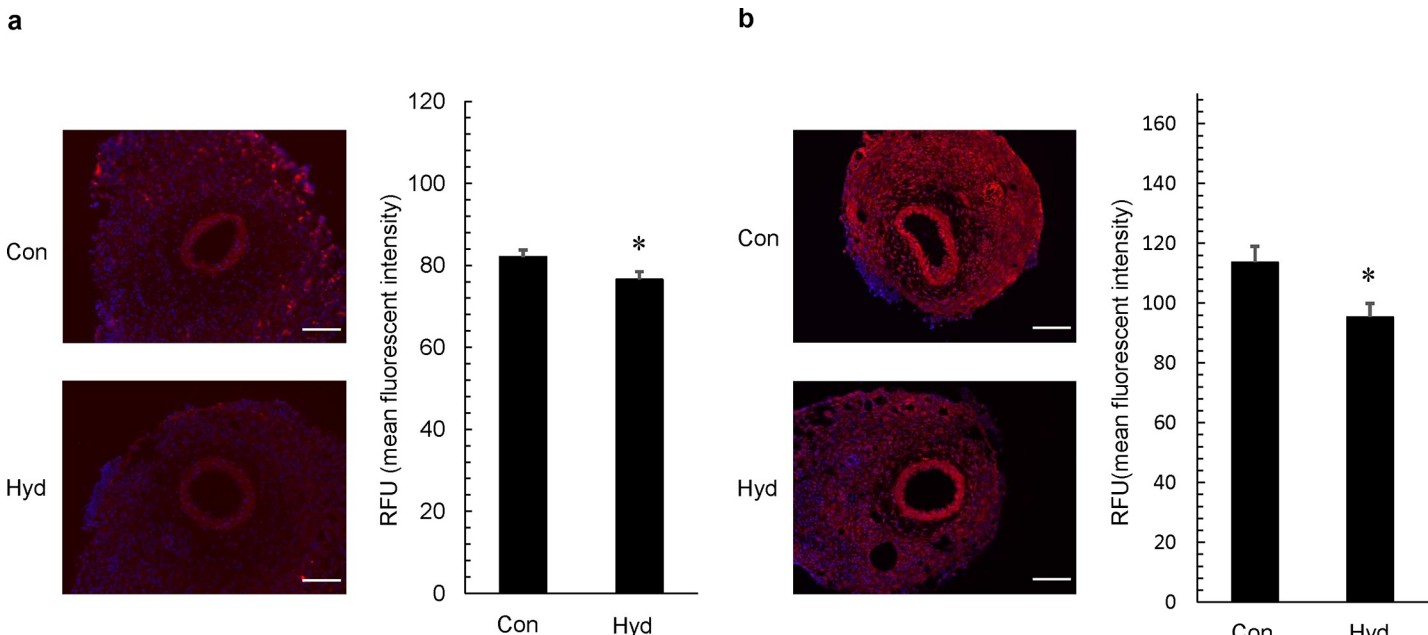

**Fig 6. Effect of hydrogen gas on DNA damage by ROS.** Tissue samples were prepared from cuffed arteries 7 days after operation. (A) Representative photos of cross-sections of injured femoral artery after immunohistochemical staining with anti-nitroguanidine antibody and fluorescence intensity in intima and media. The original photos were obtained as 8-bit images (original magnification ×200; scale bar: 100 μm), and data represent relative fluorescence units (RFU). Values are mean ±SEM (n = 21 for air group (Con), n = 21 for hydrogen group (Hyd)). $*p<0.05$ vs. Con.

makes it possible to alleviate DNA damage through downregulation of ·OH and ONOO⁻ levels.

In vascular remodeling events, ROS activate proliferation and migration of VSMC via a mitogen-activated protein kinase (MAPK) signal transduction pathway including extracellular signal-regulated kinase (ERK), p38 MAPK and c-Jun NH2-terminal kinase (JNK) [33]. In addition, MAPK signaling is involved in the production of inflammatory cytokines, and vascular remodeling events are accelerated by inflammation [34, 35]. It has also been established that MAPK signals and inflammation are augmented by cuff placement in our vascular injury model [18]. Some reports indicate that molecular hydrogen also regulates the MAPK signal transduction pathway and inflammation [9, 36]. Cardinal et al. reported that drinking hydrogen-rich water had an inhibitory effect on phosphorylation of ERK, p38 and JNK in a chronic allograft nephropathy model [36]; however, immunochemical staining with phosphorylated-ERK antibody did not show any difference in each group in this study (**S2 Fig**). Regardless of the routes of administration, molecular hydrogen has anti-inflammatory effects through inhibition of production of vascular remodeling-related inflammatory cytokines such as TNF-α and IL-1β [9]. However, there was no significant difference in the inhibitory effect of hydrogen on the expression level of F4/80 using immunohistochemical staining in our experiments (**S3 Fig**). Therefore, it is hard to conclude that the effect of our hydrogen administration system on vascular remodeling was mediated by affecting inflammatory cytokines. Further investigation is necessary to elucidate the detailed mechanism.

On the other hand, some questions were raised in this study. Inhalation as a means of administration in daily living is most likely to have the capability of constitutive administration compared with drinking or injection etc. Although our study indicated that constitutive administration of hydrogen gas at a low concentration partially attenuated vascular remodeling via reduction of oxidative stress, it remains unknown whether intermittent administration by this hydrogen administration system is effective for prevention of vascular remodeling. Indeed, it is difficult to constitutively inhale hydrogen gas in daily living, because patients are not constantly at home. Moreover, it is also necessary to investigate whether there is an inhibitory effect of a hydrogen administration system in a lifestyle-related disease model (e.g., KKAy as diabetes model mice), since CVD risk is increased by a combination of lifestyle-related diseases. Thus, further investigation is necessary.

In conclusion, our findings support the notion that long-term hydrogen gas inhalation at a safe concentration has a beneficial effect on vascular remodeling, at least due to its inhibitory effects on ROS such as ·OH and ONOO⁻, DNA damage and cell proliferation. Recently, a hydrogen-producing machine has been used safely, and constitutive hydrogen inhalation has been suggested to provide benefit involuntarily in daily life. On the other hand, drinking hydrogen-rich water has a temporary effect and the patient needs to intend to drink hydrogen-rich water. Therefore, we consider that inhalation is a more effective and natural method for administration of hydrogen. We can expect that hydrogen gas inhalation in the living environment could be useful for attenuating vascular diseases such as atherosclerosis.

## Supporting information

**S1 Checklist. The ARRIVE guidelines checklist.**
(PDF)

**S1 Table. The sequences of PCR primers.**
(DOCX)

**S1 Fig. Smooth muscle cells (α-SMA-positive cells) in neointima.** Representative photos of cross-sections of cuff (-) and cuff (+) femoral artery after immunohistochemical staining (for α-SMA). The cuff (+) femoral artery was sampled after 14 days of cuff placement. Sections were stained with the primary antibody, α-SMA antigen antibody (SIGMA, MO, USA). The methods were described the same as above in section of Immunohistochemical Staining (PCNA). Original magnification ×200 (scale bar: 30 μm).
(TIF)

**S2 Fig. Effect of hydrogen gas inhalation on expression of phosphorylated-ERK in injured femoral artery after cuff placement.** Representative photos and quantitative analysis of cross-sections of injured femoral artery after immunohistochemical staining (for phosphorylated-ERK). Sections were stained with the primary antibody, phosphorylated-ERK antigen antibody (Cell Signaling Technology, MO, USA). The methods were described the same as above in section of Immunohistochemical Staining (PCNA). Original magnification ×200 (scale bar: 30 μm). Original magnification ×600 (scale bar: 20 μm). In quantitative analysis, data represent the ratio of phosphorylated-ERK-positive area in neointima and vascular media, and values are mean ± SEM (n = 16 for air group (Con), n = 15 for hydrogen group (Hyd)). P = 0.48 vs. Con.
(TIF)

**S3 Fig. Effect of hydrogen gas inhalation on expression of F4/80 in injured femoral artery after cuff placement.** Representative photos and quantitative analysis of cross-sections of injured femoral artery after immunohistochemical staining (for F4/80). Sections were stained with the primary antibody, F4/80 antigen antibody (BMA Biomedicals, Augst, Switzerland). The methods were described the same as above in section of Immunohistochemical Staining (PCNA). Original magnification ×600 (scale bar: 20 μm). In quantitative analysis, data represent the ratio of F4/80-positive area in neointima, and values are mean ± SEM (n = 16 for air group (Con), n = 15 for hydrogen group (Hyd)). P = 0.98 vs. Con.
(TIF)

## Acknowledgments

This study was technically supported by the Division of Analytical Bio-Medicine and the Division of Laboratory Animal Research, the Advanced Research Support Center (ADRES), Ehime University.

## Author Contributions

**Data curation:** Takeshi Kiyoi, Shuang Liu, Erika Takemasa, Hirotomo Nakaoka.

**Formal analysis:** Takeshi Kiyoi, Shuang Liu.

**Funding acquisition:** Shuang Liu, Naohito Hato, Masaki Mogi.

**Methodology:** Masaki Mogi.

**Project administration:** Naohito Hato, Masaki Mogi.

**Supervision:** Naohito Hato.

**Visualization:** Takeshi Kiyoi.

**Writing – original draft:** Takeshi Kiyoi, Shuang Liu, Masaki Mogi.

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
