## [Decision Letter · Decision Letter 0]

1 Nov 2019

PONE-D-19-24002

Constitutive hydrogen inhalation prevents vascular remodeling via reduction of oxidative stress

PLOS ONE

Dear Dr. Mogi,

Thank you for submitting your manuscript to PLOS ONE. After careful consideration, we feel that it has merit but does not fully meet PLOS ONE’s publication criteria as it currently stands. Therefore, we invite you to submit a revised version of the manuscript that addresses the points raised during the review process.

We would appreciate receiving your revised manuscript by Dec 16 2019 11:59PM. To enhance the reproducibility of your results, we recommend that if applicable you deposit your laboratory protocols in protocols.io, where a protocol can be assigned its own identifier (DOI) such that it can be cited independently in the future. For instructions see: http://journals.plos.org/plosone/s/submission-guidelines#loc-laboratory-protocols

We look forward to receiving your revised manuscript.

Kind regards,

Michael Bader

Academic Editor

PLOS ONE

Journal Requirements:

2. Please complete and submit a copy of the ARRIVE Guidelines checklist, a document that aims to improve experimental reporting and reproducibility of animal studies for purposes of post-publication data analysis and reproducibility: https://www.nc3rs.org.uk/arrive-guidelines. Please include your completed checklist as a Supporting Information file. Note that if your paper is accepted for publication, this checklist will be published as part of your article.

Specifically, please ensure that you revise your methods section to include the method of euthanasia, as well as how frequently the condition of the animals was monitored.

4. Thank you for stating the following in the Financial Disclosure section:

This study was supported by JSPS KAKENHI Grant Numbers 18K08389 (S.L.) and 25462220 (M.M.), and research grants from Panasonic Corporation (https://www.panasonic.com/jp/home.html). The funders had no role in the study design, data collection and analysis, decision to publish, or preparation of the manuscript.

We note that you received funding from a commercial source: Panasonic Corporation

Reviewers' comments:

Reviewer's Responses to Questions

**Comments to the Author**

1. Is the manuscript technically sound, and do the data support the conclusions?

Reviewer #1: Partly

Reviewer #2: Yes

2. Has the statistical analysis been performed appropriately and rigorously? 

Reviewer #1: No

Reviewer #2: I Don't Know

3. Have the authors made all data underlying the findings in their manuscript fully available?

Reviewer #1: Yes

Reviewer #2: Yes

4. Is the manuscript presented in an intelligible fashion and written in standard English?

Reviewer #1: No

Reviewer #2: Yes

5. Review Comments to the Author

Reviewer #1: In this work, the authors investigated the effect of hydrogen gas inhalation to prevent vascular diseases. The authors have demonstrated that constitutive inhalation of hydrogen gas has a beneficial effect on vascular remodeling, partly due to its inhibitory effects on ROS generation, DNA damage and cell proliferation. I think this is a meaningful work drawing some interesting conclusions. But, The design of animal group and free radical signal detected is inappropriate.

Reviewer #2: Specific comments

Introduction

Line 81, 88 - How the constitutive inhalation of H2 in real life can contribute to a healthy lifestyle (from a viewpoint of medical practice).

Materials and Methods Line 98 - Compressed air containing 1.25% hydrogen gas cannot consist of O2 21% and N2 79%

Results

Please explain differences:

Line 187 and 188 - We examined the effect of hydrogen gas on neointima formation, 14 days after polyethylene cuff placement around the femoral artery….but in Material and Method you write: Line 137 - Femoral arteries that had undergone cuff placement for 7 days… or line 147 group at 7 days after cuff...or line 154- Femoral arteries that had undergone performed cuff placement for 7 days …or line 222 media of the femoral artery 7 days after cuff placement… or line 230 quantitative RT-PCR in femoral artery 7 days after… or line 231 prepared from cuffed arteries 7 days after operation… and all other figs. have 7 days

Discussion

Explain: healthy daily living in rats Line 278: remodeling aimed at healthy daily living was examined.

Line 279 What do you mean by: In this vascular injury model, NADPH oxidase activity and production of superoxide are known to be closely related,

Line 302 - Molecular hydrogen alleviates oxidative stress by acting as a radical scavenger for OH- and ONOO- in vitro -please discuss why only in vitro. Is it functioning only in vitro or in vivo as well? Are there other possibilities of antioxidative action in vivo?

Line 351 - Indeed, it is difficult to constitutively inhale hydrogen gas in daily living, because patients are not constantly at home. Please discuss if possible: Is it only reason or is it completely useless in everyday life? Is it better to drink HRW several times a day better?

6. PLOS authors have the option to publish the peer review history of their article (what does this mean?). If published, this will include your full peer review and any attached files.

Reviewer #1: No

Reviewer #2: No

---

## [Author Response · Author response to Decision Letter 0]

14 Dec 2019

Response to Editor: (MS # PONE-D-19-24002)

Thank you very much for your careful reading of our manuscript and helpful comments. In response to your comments, we have revised the manuscript carefully. Major changes are highlighted in the text. We believe this revised manuscript has been greatly improved by your constructive comments, and would be grateful if it could be re-considered for publication in PLOS ONE.

Comments:

1. When submitting your revision, we need you to address these additional requirements. Please ensure that your manuscript meets PLOS ONE's style requirements, including those for file naming.

Response:

We have checked and revised the entire manuscript accordingly.

2. Please complete and submit a copy of the ARRIVE Guidelines checklist, a document that aims to improve experimental reporting and reproducibility of animal studies for purposes of post-publication data analysis and reproducibility: Please include your completed checklist as a Supporting Information file. Note that if your paper is accepted for publication, this checklist will be published as part of your article.

Specifically, please ensure that you revise your methods section to include the method of euthanasia, as well as how frequently the condition of the animals was monitored.

Response:

We have checked the ARRIVE Guidelines and completed the ARRIVE Guidelines checklist. The document has been submitted as a Supporting Information file.

In addition, we have added the requested information to the “Animals and Treatment” section of the “Materials and Methods”.

Response:

We understand the PLOS ONE policy. We have omitted use of the phrase “data not shown”. The data are now shown in the Supporting Information file.

4. Thank you for stating the following in the Financial Disclosure section:

We note that you received funding from a commercial source: Panasonic Corporation

If there are restrictions on sharing of data and/or materials, please state these. Please note that we cannot proceed with consideration of your article until this information has been declared.

Response:

The authors have no conflict of interest including employment, consultancy, patents, products in development, marketed products, etc. from Panasonic Corporation. We have added this statement in the revision. Moreover, we understand the PLOS ONE policy. There is no conflict of interest in this research, and this fact is stated in the cover letter.

 

Response to Reviewer #1: (MS # PONE-D-19-24002)

Thank you very much for your careful reading of our manuscript and helpful comments. In response to your comments, we have revised the manuscript carefully. Major changes are highlighted in the text. We believe this revised manuscript has been greatly improved by your constructive comments, and would be grateful if it could be re-considered for publication in PLOS ONE.

Comments:

In this work, the authors investigated the effect of hydrogen gas inhalation to prevent vascular diseases. The authors have demonstrated that constitutive inhalation of hydrogen gas has a beneficial effect on vascular remodeling, partly due to its inhibitory effects on ROS generation, DNA damage and cell proliferation. I think this is a meaningful work drawing some interesting conclusions. But, the design of animal group and free radical signal detected is inappropriate.

Response: 

We appreciate your valuable comments. According to the comments, we have added more details on the design of animal groups and free radical signal detection in the “Materials and Methods” of the revised manuscript, as described in the response to the Editor’s comments.

 

Response to Reviewer #2: (MS # PONE-D-19-24002)

Thank you very much for your careful reading of our manuscript and helpful comments. In response to your comments, we have revised the manuscript carefully. Major changes are highlighted in the text. We believe this revised manuscript has been greatly improved by your constructive comments, and would be grateful if it could be re-considered for publication in PLOS ONE.

Specific comments

Introduction

Line 81, 88 - How the constitutive inhalation of H2 in real life can contribute to a healthy lifestyle (from a viewpoint of medical practice).

Response:

We appreciate your constructive comment. We have revised the manuscript as follows:

Introduction (Line 62)

“It was suggested that molecular hydrogen prevents vascular remodeling in animal models such as ischemia and reperfusion (I/R) injury, vein grafting, carotid balloon injury and cerebral vasospasm in subarachnoid hemorrhage via reduction of oxidative stress (12-15).”

Introduction (Line 74)

“In this study, we have focused on the effects of hydrogen gas inhalation on CVD as a lifestyle intervention. CVD is induced by lifestyle-related disease with chronic/persistent oxidative stress; that is, the constitutive inhalation of molecular hydrogen in real life contributes to reducing chronic/persistent oxidative stress and has the potential to prevent CVD.”

Materials and Methods

Line 98 - Compressed air containing 1.25% hydrogen gas cannot consist of O2 21% and N2 79%

Response:

We appreciate your comment. We have revised the manuscript as follows:

Materials and Methods (Line 85)

“Compressed hydrogen gas (O2 21%, N2 77.7%, hydrogen 1.3%) or compressed air (O2 21%, N2 79%) flowed continuously at 0.4 L/min.”

Results

Please explain differences:

Line 187 and 188 - We examined the effect of hydrogen gas on neointima formation, 14 days after polyethylene cuff placement around the femoral artery….but in Material and Method you write: Line 137 - Femoral arteries that had undergone cuff placement for 7 days… or line 147 group at 7 days after cuff...or line 154- Femoral arteries that had undergone performed cuff placement for 7 days …or line 222 media of the femoral artery 7 days after cuff placement… or line 230 quantitative RT-PCR in femoral artery 7 days after… or line 231 prepared from cuffed arteries 7 days after operation… and all other figs. have 7 days

Response:

We appreciate your comments. We previously reported that neointima is formed via increasing oxidative stress and NADPH oxidase. Neointima formation in the femoral artery 14 days after cuff placement was reflected by increased oxidative stress and mRNA levels of NADPH oxidase subunits at 7 days after cuff placement. Therefore, we conducted experiments of quantitative RT-PCR and detection of ROS at 7 days after cuff placement. Therefore, we assessed ROS production and mRNA expression by RT-PCR at 7 days after cuff placement. The following references have been helpful.

17. Chisaka T, Mogi M, Nakaoka H, Kan-No H, Tsukuda K, Wang XL, Bai HY, Shan BS, Kukida M, Iwanami J, Higaki T, Ishii E, Horiuchi M. Low-protein diet-induced fetal growth restriction leads to exaggerated proliferative response to vascular injury in postnatal life. Am J Hypertens. 2016;29(1):54-62. 

18. Ohnishi A, Asayama R, Mogi M, Nakaoka H, Kan-No H, Tsukuda K, Chisaka T, Wang XL, Bai HY, Shan BS, Kukida M, Iwanami J, Horiuchi M. Drinking citrus fruit juice inhibits vascular remodeling in cuff-induced vascular injury mouse model. PLoS One. 2015;10(2):e0117616.

Discussion

Explain: healthy daily living in rats Line 278: remodeling aimed at healthy daily living was examined.

Response:

We appreciate your comment. We have revised this as follows:

Discussion (Line 272)

On the other hand, this study on the effects of hydrogen gas inhalation on CVD focused on lifestyle intervention. In the present study, considering the daily life of humans, the preventive effect of constitutive administration of hydrogen gas on vascular remodeling was examined in a mouse cuff injury model. 

Line 279 What do you mean by: In this vascular injury model, NADPH oxidase activity and production of superoxide are known to be closely related,

Response:

We appreciate your comment. We have revised this as follows:

Discussion (Line 277)

“In this vascular injury model, it is known that superoxide resulting from increased NADPH oxidase activity promotes VSMC proliferation and neointima formation (21).”

Line 302 - Molecular hydrogen alleviates oxidative stress by acting as a radical scavenger for OH- and ONOO- in vitro -please discuss why only in vitro. Is it functioning only in vitro or in vivo as well? Are there other possibilities of antioxidative action in vivo?

Response:

We appreciate your comment. There are some reports that molecular hydrogen acts in vivo as well as in vitro. We have described two reports with respect to in vivo study.

Discussion (Line 280)

“Zhang et al. indicated that intraperitoneal injection of hydrogen-rich medium produced a decrease in expression of NADPH oxidase in an isoproterenol (ISO)-induced cardiac hypertrophy model rat (24). Qin et al. reported that injection of hydrogen-rich saline reduced superoxide and prevented VSMC proliferation and migration in a rat carotid balloon injury model (13). These reports suggest that molecular hydrogen can prevent CVD by downregulating NADPH activity and superoxide production. However, molecular hydrogen does not act as a radical scavenger for other ROS such as superoxide, hydrogen peroxide etc. in vitro (6). This is an unclear point regarding the mechanism of action of molecular hydrogen on biological activity. In the present results, inhalation of hydrogen gas downregulated the expression of NADPH oxidase, NOX1, but did not affect the expression levels of NADPH oxidase subunits such as p40phox and p47phox in the femoral artery 7 days after cuff placement. "

Line 351 - Indeed, it is difficult to constitutively inhale hydrogen gas in daily living, because patients are not constantly at home. Please discuss if possible: Is it only reason or is it completely useless in everyday life? Is it better to drink HRW several times a day better?

Response:

We appreciate your comments. Recently, a hydrogen-producing machine has been used safely, and constitutive hydrogen inhalation has been suggested to provide benefit involuntarily in daily living. On the other hand, drinking HRW has a temporary effect and the patient needs to intend to drink HRW. Therefore, we consider that inhalation is the most effective and natural method for administration of hydrogen. We have revised the manuscript as follows:

Discussion (Line 360)

“Recently, a hydrogen-producing machine has been used safely, and constitutive hydrogen inhalation has been suggested to provide benefit involuntarily in daily living. On the other hand, drinking hydrogen-rich water has a temporary effect and the patient needs to intend to drink hydrogen-rich water. Therefore, we consider that inhalation is a more effective and natural method for administration of hydrogen. We can expect that hydrogen gas inhalation in the living environment could be useful for attenuating vascular diseases such as atherosclerosis.”

---

## [Editor Report · Decision Letter 1]

23 Dec 2019

Constitutive hydrogen inhalation prevents vascular remodeling via reduction of oxidative stress

PONE-D-19-24002R1

Dear Dr. Mogi,

We are pleased to inform you that your manuscript has been judged scientifically suitable for publication and will be formally accepted for publication once it complies with all outstanding technical requirements.

With kind regards,

Michael Bader

Academic Editor

PLOS ONE
---

## [Editor Report · Acceptance letter]

7 Jan 2020

PONE-D-19-24002R1 

Constitutive hydrogen inhalation prevents vascular remodeling via reduction of oxidative stress 

Dear Dr. Mogi:

I am pleased to inform you that your manuscript has been deemed suitable for publication in PLOS ONE. Congratulations! Your manuscript is now with our production department. 

With kind regards,

on behalf of

Prof. Michael Bader 

Academic Editor

PLOS ONE